# Phase II, double blind, placebo controlled, multi-site study to evaluate the safety, feasibility and desirability of conducting a phase III study of anamorelin for anorexia in people with small cell lung cancer: A study protocol (LUANA trial)

Mariana S. Sousa[1,2], Peter Martin[3,4], Miriam J. Johnson[5], Michael Lind[5], Matthew Maddocks[6], Alex Bullock[5], Meera Agar[1,7,8], Sungwon Chang[1], Slavica Kochovska[1,9], Irina Kinchin[1,10], Deidre Morgan[11], Belinda Fazekas[1], Valentina Razmovski-Naumovski[1,8], Jessica T. Lee[1,12,13], Malinda Itchins[14,15], Victoria Bray[7], David C. Currow[1,9]*

1 Faculty of Health, Improving Palliative, Aged and Chronic Care through Clinical Research and Translation (IMPACCT), University of Technology Sydney, Ultimo, New South Wales, Australia, 2 School of Nursing and Midwifery, Western Sydney University, Penrith, New South Wales, Australia, 3 School of Medicine, Deakin University, Geelong, Victoria, Australia, 4 Palliative Care, Barwon Health, Geelong, Victoria, Australia, 5 Wolfson Palliative Care Research Centre, Hull York Medical School, University of Hull, Hull, England, 6 Cicely Saunders Institute of Palliative Care, Policy and Rehabilitation, King's College London, Denmark Hill, London, United Kingdom, 7 Department of Medical Oncology, Liverpool Hospital, Liverpool, New South Wales, Australia, 8 South West Sydney Clinical Campuses, University of New South Wales Sydney, Kensington, Australia, 9 Faculty of Science, Medicine and Health, University of Wollongong, Wollongong, New South Wales, Australia, 10 Trinity College Dublin, The University of Dublin, Dublin, Ireland, 11 RePaDD, Flinders University, Adelaide, South Australia, Australia, 12 Concord Centre for Palliative Care, Concord Repatriation General Hospital, Concord West, New South Wales, Australia, 13 Concord Clinical school, University of Sydney, Camperdown, New South Wales, Australia, 14 Royal North Shore Hospital, St Leonards, New South Wales, Australia, 15 Faculty of Medicine and Health, University of Sydney, Camperdown, New South Wales, Australia

* dcurrow@uow.edu.au

## Abstract

Anorexia is experienced by most people with lung cancer during the course of their disease and treatment. Anorexia reduces response to chemotherapy and the ability of patients to cope with, and complete their treatment leading to greater morbidity, poorer prognosis and outcomes. Despite the significant importance of cancer-related anorexia, current therapies are limited, have marginal benefits and unwarranted side effects. In this multi-site, randomised, double blind, placebo controlled, phase II trial, participants will be randomly assigned (1:1) to receive once-daily oral dosing of 100mg of anamorelin HCl or matched placebo for 12 weeks. Participants can then opt into an extension phase to receive blinded intervention for another 12 weeks (weeks 13–24) at the same dose and frequency. Adults (≥18 years) with small cell lung cancer (SCLC); newly diagnosed with planned systemic therapy OR with first recurrence of disease following a documented disease-free interval ≥6 months, AND with anorexia (i.e., ≤ 37 points on the 12-item Functional Assessment of Anorexia Cachexia

**Data Availability Statement:** Any requirements for data sharing during the process of publishing results in a peer-reviewed publication will comply with the data privacy and Human Research Ethics Committee requirements. Publications in scientific journals is planned as well as presentation of outcomes at national and international scientific meetings.

**Funding:** This study is sponsored by the University of Technology Sydney and funded by the Helsinn Group including the provision of study medication and matching placebo. The funders were consulted on study design but have no role on data collection and analysis, decision to publish, or preparation of the manuscript.

**Competing interests:** Prof David Currow is a paid consultant to Helsinn and a member of their Advisory Board. This does not alter our adherence to PLOS ONE policies on sharing data and materials. All the other Authors declare that they have no conflict of interest.

Treatment (FAACT A/CS) scale) will be invited to participate. Primary outcomes are safety, desirability and feasibility outcomes related to participant recruitment, adherence to interventions, and completion of study tools to inform the design of a robust Phase III effectiveness trial. Secondary outcomes are the effects of study interventions on body weight and composition, functional status, nutritional intake, biochemistry, fatigue, harms, survival and quality of life. Primary and secondary efficacy analysis will be conducted at 12 weeks. Additional exploratory efficacy and safety analyses will also be conducted at 24 weeks to collect data over longer treatment duration. The feasibility of economic evaluations in Phase III trial will be assessed, including the indicative costs and benefits of anamorelin for SCLC to the healthcare system and society, the choice of methods for data collection and the future evaluation design. **Trial registration.** The trial has been registered with the Australian New Zealand Clinical Trials Registry [ACTRN12622000129785] and approved by the South Western Sydney Local Health District Human Research Ethics Committee [2021/ETH11339]. https://clin.larvol.com/trial-detail/ACTRN12622000129785.

## Introduction

Cancer-related anorexia (appetite loss) remains one of the most prevalent and bothersome clinical problems experienced by people with cancer during and after treatment [1]. It is estimated that over 50% of people present with anorexia at the time of a cancer diagnosis [1] and is an independent negative prognostic factor for survival [2].

Cancer anorexia is a major contributor to weight loss and adversely influences nutritional status in advanced cancer [1]. People with anorexia generally have reduced physical capacity, often reporting being too weak to perform daily tasks, thus limiting independence, and time with their family and friends [3]. Anorexia also reduces the effectiveness of chemotherapy and the ability of people to cope with, and complete their treatment (on-time and on-dose) [4]. Cancer anorexia has also been linked to lower social functioning and depression, with profound impacts on each individual's psychological wellbeing and social life, all of which may reduce perceived quality of life (QoL) [1]. Despite the significant impact of cancer-related anorexia, effective and long-term treatments are lacking. The treatment of malignancy-associated anorexia remains an area of ongoing clinical need.

Lung cancer is one of the most commonly occurring cancers representing over 12% of the global cancer burden and continues to be the leading cause of cancer-related deaths [5]. There are two main sub-types of lung cancer: non-small cell lung cancer (NSCLC) and small cell lung cancer (SCLC), with the latter associated with greater mortality. People with SCLC have a very high prevalence of distressing appetite loss [6], and therefore, are investigated in this study. Approximately two thirds of people with advanced SCLC experience anorexia [7,8].

Ghrelin is an endogenous stimulator of appetite which specifically targets the growth hormone secretagogue receptor (GHSR-1a). Similar to ghrelin, anamorelin–a ghrelin agonist—increases GH, insulin-like growth factor-1 and insulin-like growth factor binding protein-3 thus producing significant appetite stimulation. Two international, phase III trials (ROMANA 1 [HT-ANAM-301] and ROMANA 2 [HT-ANAM-302]) [9,10] and an associated extension study (ROMANA 3 [HT-ANAM-303]) [11] tested anamorelin for cancer anorexia and cachexia in people with inoperable stage III and IV NSCLC. These placebo-controlled trials, with more than 900 participants, demonstrated that anamorelin safely reversed muscle loss, augmented body weight with sustainably improved appetite and was well tolerated. However,

the trials failed to meet functional co-primary endpoints (improved hand-grip strength), food intake was not recorded and it is not known if the improvement in anorexia translated into improved nutritional intake or better oncologic outcomes [12]. Therefore, despite promising findings, current data on the safety and efficacy of the medicinal product was not sufficiently demonstrated to inform a registration of anamorelin by the Food and Drug Administration (FDA) in the USA or the European Medicines Agency (EMA) for use in cancer cachexia. A world first, anamorelin was recently approved in Japan for cancer cachexia-related appetite and weight loss in people with NSCLC, gastric, pancreatic and colorectal cancers [13]. By contrast to ROMANA studies with cachexia as the primary end-point, this study will focus on anorexia, and the ability to tolerate anti-cancer therapies.

This phase II trial aims to determine the safety, feasibility and desirability (sufficient signal in the efficacy evaluation) of a fully powered placebo-controlled phase III study of anamorelin HCl (100 mg) for treating anorexia in people with SCLC. Feasibility in this trial is related to participant recruitment, adherence to, and completion of, study tools and interventions. Efficacy is determined by change in anorexia-related symptoms from baseline to week 12 and completion of any planned chemotherapy and/or radiation therapy.

## Material and methods

### Study design and protocol registration

This is a phase II, double blind, placebo controlled, parallel arm, fixed dose, multi-site study, prospectively registered at the Australian New Zealand Clinical Trials Registry [**ACTRN12622000129785**].

This protocol adheres to the SPIRIT (Standard Protocol Items: Recommendations for Interventional Trials) guidelines for trial protocol development and the results will be reported following the CONSORT (CONsolidated Standards of Reporting Trials) reporting framework (http://www.consort-statement.org/). The study was reviewed and approved by the South Western Sydney Local Health District Human Research Ethics Committee [**2021/ETH11339**] on February 09, 2022.

### Recruitment and eligibility criteria

Participants will be recruited using advertisements in waiting rooms and clinician referrals from oncology clinics at six hospitals across Australia. Adults (aged ≥18 years) with a confirmed diagnosis of small cell lung cancer (SCLC) are referred to the study nurse at each participating centre and assessed for eligibility. People who meet all the inclusion criteria and none of the exclusion criteria are eligible to enter this study (Table 1). All participants will be required to provide written, informed consent prior to participation.

Any participant who has consented to join the study and meets the eligibility criteria will be asked to nominate their primary caregiver where present, and asked for permission to approach them regarding the study. The caregiver will be asked to provide informed consent and if they would be prepared to complete the caregiver unmet needs and QoL assessments. A primary caregiver is defined as a spouse/relative, partner or friend (aged ≥18 years) who provides care for the person with SCLC and not someone who has been employed to care for them. Consent by the caregiver, or otherwise, will not have any effect on the continued participation of the person they care for in the main trial.

### Sample size

For a feasibility study, no formal sample size calculation is required [14]. It has been suggested, however, that in feasibility studies, a sample size between 24 and 55 is acceptable [15,16]. The

**Table 1. List of inclusion and exclusion criteria.**

**INCLUSION CRITERIA**
**To be eligible, subjects must meet ALL of the following inclusion criteria.**

- ≥18 years of age.
- Documented histologic or cytologic diagnosis of small cell lung cancer (limited–one lung and/or nearby lymph nodes; or extensive disease–extends beyond single lung, and extended to other lymph nodes or other parts of the body).
- Newly diagnosed with small cell lung cancer with planned systemic therapy* OR first recurrence of disease following successful treatment with a documented disease-free interval of ≥ 6 months.
- ≤ 37 points on the 12-item Functional Assessment of Anorexia Cachexia Treatment (FAACT A/CS) scale.
- Australia-modified Karnofsky Performance status ≥50 at screening.
- Adequate hepatic function [AST (SGOT) and ALT (SGPT) ≤ 5 x ULN].
- Adequate renal function (calculated creatinine clearance > 20 mL/minute).
- English-speaking (or have an interpreter available).
- The participant must be willing and able to provide written informed consent, and comply with the protocol tests and procedures.
- Female participants shall be**:
  - of non-childbearing potential; OR
  - of childbearing potential using reliable contraceptive measures AND having a negative urine pregnancy at baseline prior to first dose of investigational product.

**EXCLUSION CRITERIA**
**Subjects who meet ANY of the following exclusion criteria must be excluded from the study.**

- Women who are pregnant OR breastfeeding.
- Pathology and causes that may impede food intake, as determined by the Investigator. These causes may include but are not limited to:
  - Grade 3 or 4 oral mucositis;
  - Grade 3 or 4 GI disorders (nausea, vomiting, diarrhea, and constipation); OR
- mechanical obstructions making the person unable to eat.
- Having undergone major surgery (central venous access placement and tumour biopsies are not considered major surgery) within 4 weeks prior to randomisation. Potential participants must be recovered from acute effects of surgery prior to screening. Participants should not have a current treatment plan to undergo major surgical procedures during the treatment period.
- Currently taking androgenic compounds (including but not limited to testosterone, testosterone-like agents, oxandrolone, megestrol acetate, methylphenidate, corticosteroids), olanzapine, prokinetics (including metoclopramide), dronabinol or medical marijuana (medical cannabis) or any other prescription medication or off-label products intended to increase appetite or treat unintentional weight loss [e.g., melatonin, nabilone, delta-9-tetrahydrocannabinol (THC) and cannabidiol (CBD)]–With the exception when any of these medications are administered (short-term) as part of routine chemotherapy/ radiation therapy standard protocols.
- On mirtazapine in the previous four weeks.
- Pleural effusion requiring thoracentesis.
- Pericardial effusion requiring drainage.
- Oedema requiring regular diuretics.
- Ascites requiring drainage.
- Uncontrolled or significant cardiovascular disease, including but not limited to:
  - history of myocardial infarction within the past 3 months;
  - A-V block of second or third degree (but eligible if currently has a pacemaker–with the exception that BIA will not be performed if the person has a pacemaker, due to minor electrical current);
  - unstable angina;
  - congestive heart failure within the past 3 months, if defined as New York Heart Association (NYHA) class III-IV;
  - any history of clinically significant ventricular arrhythmias (such as ventricular tachycardia, ventricular fibrillation, Wolff-Parkinson-White (WPW) syndrome, or torsade de pointes);
  - uncontrolled hypertension (blood pressure >160 mmHg systolic and >100 mmHg diastolic);
  - heart rate < 50 beats per minute on pre-entry electrocardiogram and participant is symptomatic.
- Taking regular medications that may prolong the PR or QRS interval durations, such as any of the antiarrhythmic medications Class I [fast sodium (Na) channel blockers (e.g., quinidine, disopyramide, procainamide, lidocaine, phenytoin, flecainide, propafenone)].
- Unable to swallow oral tablets.
- Severe gastrointestinal disease (including oesophagitis, gastritis, malabsorption).
- History of a gastrectomy.
- Recent history of radiotherapy of oesophagus area.
- Diabetes mellitus with secondary organ dysfunction (coronary heart disease, previous stroke, renal insufficiency), or poorly controlled diabetes (patients with glycosylated haemoglobin– HbA1c >7% or hyperglycaemia–measured as a fasting blood glucose >7mmol/L or a random blood glucose >11mmol/L) despite receiving clinic-based diabetes care.
- Diagnosis of anorexia caused by other reasons, as determined by the investigator such as:
  - advanced AIDS;
  - heart failure;
  - uncontrolled thyroid disease.
- Receiving strong CYP3A4 inhibitors (including clarithromycin, erythromycin, diltiazem, itraconazole, ketoconazole, ritonavir, verapamil, etc.) within 14 days of randomization.

*(Continued)*

**Table 1.** (Continued)

- Currently receiving tube feedings or parenteral nutrition (either total or partial).
- Current use of excessive alcohol or illicit drugs.
- Any condition, including the presence of laboratory abnormalities, which in the Investigator's opinion, places the potential participant at unacceptable risk if he/she were to participate in the study or confounds the ability to interpret study data.
- Enrolment in a previous study with anamorelin HCl or previous exposure to anamorelin HCl.
- Actively receiving a concurrent investigational agent or having received an investigational agent within 28 days of Day 1.
- Cognitive impairment (Short Blessed Test (SBT) score ≥10).

\*Notes:

Planned therapy includes the period from having a defined treatment schedule and study medication starting up to one day prior to day one of cycle 2 of chemotherapy. Under no circumstances should the study delay the routine treatment for SCLC.

\*\*Notes:

I) Female participants of non-childbearing potential are defined as being post-menopausal for at least 1 year; or having documented surgical sterilization or hysterectomy at least 3 months before study participation.

II) Reliable contraceptive measures include implants, injectables, combined oral contraceptives, intrauterine devices, vasectomized partner or complete (long term) sexual abstinence.

proposed sample size of 50 participants (25 per arm), will provide sufficient data to answer feasibility questions. This target should allow for sufficient information on the suitability and sensitivity of the outcome measures for use with this population and the standard deviations (SDs) of the measures to inform a sample size calculation for a future larger scale phase III trial.

## Randomisation and allocation concealment

Randomisation will occur after eligibility is confirmed, consent provided and baseline assessments are collected. Eligible participants will be randomised to receive either anamorelin or matching placebo in a 1:1 ratio. On notification of a participant, the delegated pharmacist will email the Central Randomisation Service providing the required registration details and receive the treatment allocation from a reply email. This allocation code, date of request, preparation, and dispensing will be recorded in a log maintained by the pharmacist and supplied to the central registry on each randomisation. Block randomisation of 4 or 8 will occur in each stratum at each site and will ensure even allocation to each code. Central randomisation will be stratified by limited (one lung and/or nearby lymph nodes) versus extensive disease (extends beyond single lung, and to other lymph nodes or other parts of the body) and by baseline score of the 5-item Anorexia Symptom Scale Domain from the 12-item Functional Assessment of Anorexia Cachexia Treatment (FAACT A/CS) scale (≤10 vs >10) [17].

## Blinding

This study is a double-blind design whereby treatment allocation will not be disclosed to participants, study staff, treating clinicians or investigators. The drug manufacturer ensures study drug and placebo are identical in appearance, smell and taste. All medication will be dispensed in blinded packaging on a *per participant* basis according to the treatment allocation derived from the randomisation schedule. The code will only be broken in situations where knowledge of the allocation will have immediate consequences for clinical decision making in consultation with the Lead Investigator (DCC).

## Intervention

**Description.** The intervention will be delivered as one blinded tablet daily: one anamorelin HCl 100mg tablet or one matching placebo tablet (Fig 1). Participants will be supplied with

| | STUDY PERIOD | | | | | | | |
| --- | --- | --- | --- | --- | --- | --- | --- | --- |
| | Enrolment | Allocation | Post-allocation | | | | | Close-out |
| | | | | | | Extension | | |
| **TIMEPOINT** | -t₁ | 0 | t₁ _Week 4_ | t₂ _Week 8_ | t₃ _Week 12_ | t₄ _Week 18_ | t₅ _Week 24_ | _Follow-up_[a] |
| **ENROLMENT:** | | | | | | | | |
| **Eligibility screen** | X | | | | | | | |
| **Informed consent** | X | | | | | | | |
| _Demographic, clinical information, PE, AKPS, Vitals, SBT, ECG, chemistry, pregnancy test, FAACT A/CS_ | X | | | | | | | |
| **Allocation** | | X | | | | | | |
| **INTERVENTIONS:** | | | | | | | | |
| _**Anamorelin HCl 100mg**_ | | | ←————————————→ | | | | | |
| _**Placebo**_ | | | ←————————————→ | | | | | |
| _**Standard physical activity and dietary advice leaflets¹**_ | | X | | | | | | |
| **ASSESSMENTS: Patients** | | | | | | | | |
| _**Baseline:** AKPS, CRP, IL-6, pregnancy test, FAACT A/CS, SCLC treatment regimen and dose or interval adjustments, clinical information, harms, survival, height, weight (BMI), historical weight loss, BIA, DEXA, CT^, FACIT-F, SQUASH, CGI-S, EQ-5D-5L, ICECAP-A², Diary, TUG_ | | X | | | | | | |
| _**Outcomes:** FAACT A/CS, SCLC treatment regimen and dose or interval adjustments, SQUASH, Diary, TUG, CGI-S, CGI-I, CT^, compliance, weight, harms, survival, PE, AKPS, Vitals,_ | | | X | X | X | X | X | |
| _ECG_ | | | X | X | X | | | |
| _Chemistry, CRP, IL-6*_ | | | X | | X* | | X | |
| _BIA, DEXA_ | | | | | X* | | X | |
| _FACIT-F, EQ-5D-5L, ICECAP-A²_ | | | | X | X | | X | |
| _Response to therapy, adverse events, acceptability of study and proxy_ | | | | | X | | X | |
| _GIC_ | | | X | X | X | | X | |
| _**Follow-up:** response to therapy, adverse events, SCLC treatment regimen and dose or interval adjustments, harms, survival, clinical information_ | | | | | | | | X |
| _**Caregivers** EQ-5D-5L, CES_ | | X | | | X*§ | | X§ | |
| _Diary_ | | X | X | X | X*§ | X | X§ | |

**Fig 1. Schedule of enrollment, intervention and assessment.** PE, Physical examination; AKPS, Australia-modified Karnofsky Performance Status; SBT, Short Blessed Test; ECG, Electrocardiogram; FAACT, Functional Assessment of Anorexia Cachexia Treatment; CPR, C-reactive protein; IL-6, Interleukin 6; SCLC, small cell lung cancer; BMI, Body Mass Index; BIA, Bio-electrical impedance); DEXA, dual x-ray absorptiometry; CT, computed tomography; FACIT-F, Functional Assessment of Chronic Illness Therapy-Fatigue; SQUASH, Short Questionnaire to Assess Health Enhancing Physical Activity; CGI-S, Clinical Global Impression of Severity; EQ-5D-5L, EuroQol 5-level

5-Dimensions; ICECAP-A, ICEpop CAPability Measure for Adults; TUG–Timed Up and Go test; CGI-I, Clinical Global Impression of Improvement; GIC, Global Impression of Change. [a] First telephone follow-up visit will occur 7 days after the Week 12 or Week 24 (if did not elect to enter extension) visit or the early termination visit due to withdrawal if treatment was ceased early. Subsequent telephone follow-up visits will occur every 7 days thereafter for a period of 4 weeks. There will be 4 follow-up visits in total. [1] Standard physical activity and dietary advice leaflets will be provided to both arms at baseline. [2] ICECAP-A will be administered after the EQ-5D-5L questionnaire.[^] All electronically stored data from chest CT scans whenever they are done clinically will be used for evaluation of change in muscle mass at the thoracic (T4) level. [*]Only if people elect NOT to enter the extension. [§] Or withdrawal.

study drug within two business days after randomisation and at Week 13 (optional extension phase). Day 1 of study intervention will initiate as soon as practicable; ideally prior to cycle 1 of chemotherapy and no later than one day prior to the scheduled date of cycle 2 of chemotherapy. If chemotherapy is not planned, Day 1 should occur within three weeks of screening visit. The tablets are to be taken orally, once daily before breakfast in mornings until Week 12 or up to week 24, if a participant opts in for blinded continuation in the optional extension phase (weeks 13–24). Water is permitted prior to and with study drug. If participants have missed any doses, irrespective of the number of doses missed, they will be instructed to not catch up the missed doses if the missed dose was more than 8 hours earlier, but to continue with the next day of treatment.

At baseline, all participants will be provided with, standard physical activity and dietary advice leaflets developed by the investigator team as part of a multi-modal intervention. All participants will receive the same information delivered in the same way. All other therapies will continue as prescribed by the treating physician(s). Compliance will be assessed at weeks 4, 8 and 12 (plus 18 and 24 for those in the extension phase) using participant record of self-administration (i.e. participant diary) and counting of the remaining tablets at treatment cessation, if the information has not been recorded in the participant diary (Fig 2).

**Criteria for cessation of study intervention.** Treatment may be ceased for one of the following reasons: (1) a participant withdraws their consent for continued administration of the study medication; (2) a participant withdraws their consent from study and all protocol-specified assessments and follow-up procedures; (3) reached primary endpoint; (4) at participant request; (5) insertion of tube feeding or parenteral nutrition; (6) adverse events of grade 3 not resolving with appropriate treatment, or any grade 4 or 5 toxicities (defined by National Cancer Institute Common Criteria for Adverse Events; NCI-CTCAE version 5.0) making it unsafe for the participant to continue in the study; (7) participants who, in the opinion of the Investigator, are not well enough to continue the study (specific reasons for cessation need to be documented); (8) the participant is lost to follow-up; (9) the participant dies; (10) study termination by the Sponsor, the Regulatory Authorities or the Ethics Committee; (11) it is inappropriate to continue the study medicine for whatever reason. If a participant prematurely discontinues treatment with anamorelin/placebo at any time prior to completing the 12-week visit, they will be asked to complete protocol-specified assessments up to Week 12 ("retrieved dropout" approach). If participants discontinue study participation at any time, they will be asked if they can participate in one final clinical visit and whether they would participate in a follow-up telephone call seven days later.

**Standard care.** The placebo arm will be identical in terms of administration and opportunities to enter the extension phase.

**Intercurrent care.** All routine medications and therapies well continue as prescribed by the caring physician.

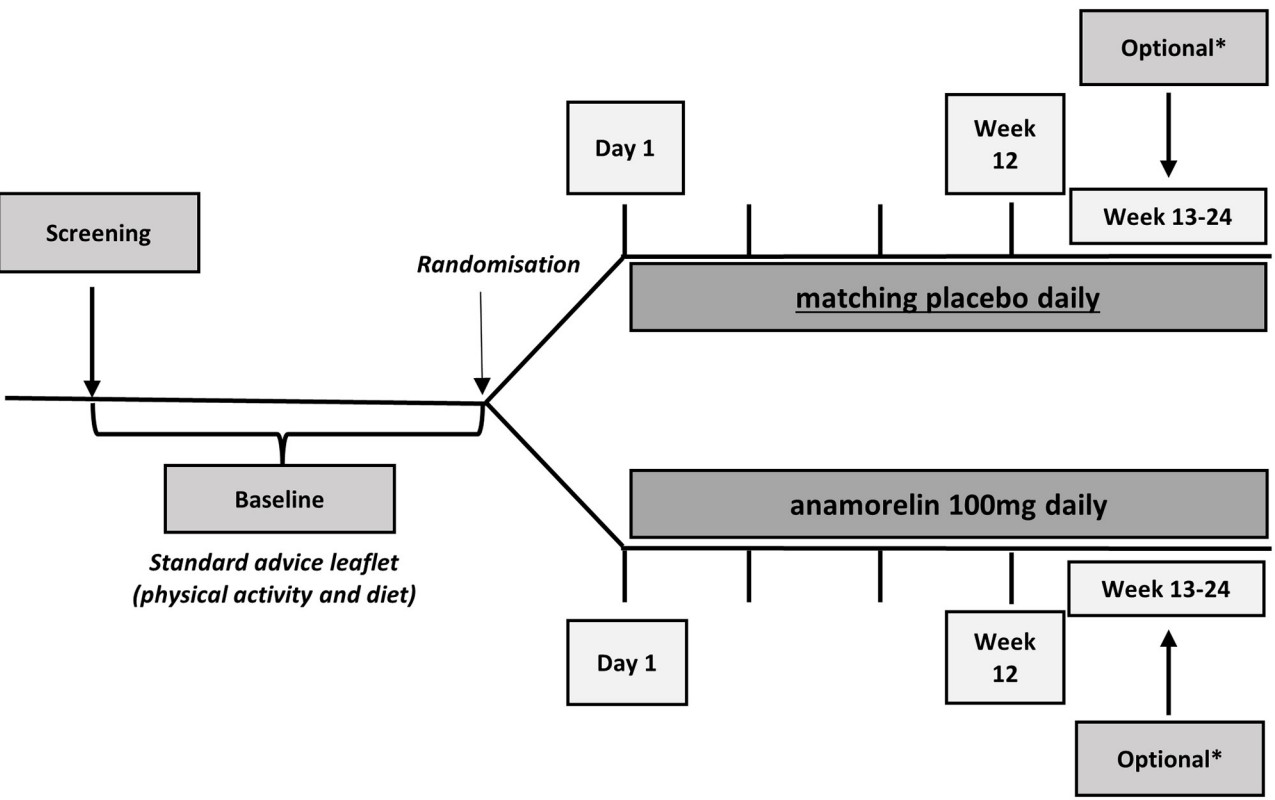

**Fig 2. Study diagram.** *The optional extension is entirely based on participants' preference for blinded continuation for weeks 13–24.

## Data collection and outcome measures

Participants will visit the study centre at baseline, Weeks 4, 8 and 12 (and 18 and 24 for those in the extension phase). During the visits, the study nurse will take the measures and assessments (Fig 1) and record the visit in the Data Collection Worksheet for that time point. The study period will include 28 days after medication cessation (4 weekly follow-up period) or until death (whichever timeframe is shorter).

Feasibility parameters will be assessed using an indicative traffic light system (green, orange red) whereby green (go) suggests that the criteria have been met and a phase III study should proceed, orange (amend) indicates that some design changes should be made prior to the larger trial and red (stop) indicates that a larger trial should not proceed. Feasibility in this trial is related to:

- recruitment rate, defined as an average of one participant per site every six weeks–plan to recruit 50 participants;

- proportion of participants with complete primary endpoint data–>70% (green), 60–70% (orange), <60% (red);

- adherence to investigational medication–>70% (green), 60–70% (orange), <60% (red);

- adherence to physical activity advice, whereby increase in physical activity will be reviewed over the study period–>40% (green), 30–40% (orange), <30% (red);

- adherence to dietary advice, whereby increase in macronutrients intake will be reviewed over the study period–>40% (green), 30–40% (orange), <30% (red).

The study will be considered desirable if there is sufficient signal in the efficacy evaluation to suggest that an adequately powered study could deliver a clinically meaningful difference between groups (where the primary measure has a pre-defined minimum clinically important difference), feasibility end-points are met and participants find the study acceptable.

Primary efficacy parameters will include (1) change in the 5-item Anorexia Symptom Scale Domain from the 12-item Functional Assessment of Anorexia Cachexia Treatment (FAACT, anorexia/cachexia–A/CS) scale by arm from baseline to week 12 and (2) rates of *on time*, *on dose* completion of any administered chemotherapy and/or radiotherapy. Secondary preliminary efficacy, harms and health economic outcomes will be assessed by a number of methods and validated tools according to Table 2. In summary, secondary parameters will include: (1) change in body weight; (2) change in lean body mass and muscle mass at thoracic (T4) level; (3) functional status; (4) performance-based functional mobility and physical activity; (5) nutritional intake; (6) biochemical changes including in pre-albumin, CRP and IL-6; (7) survival (overall and cancer-specific); (8) participants' and clinicians' rating of overall improvement; (9) quality of life; (10) chronic illness therapy-fatigue; (11) rates of unplanned health care contact (including inpatient/outpatient attendance and community care such as emergency department contact, unplanned primary care or specialist contact); (12) rates of hospitalisations and length of stay; (13) adverse events from chemotherapy (e.g., nausea, vomiting, anorexia, fatigue); (14) rates of febrile neutropenia and (15) harms.

Safety assessments are made at all contacts with participants, before efficacy assessments and consist of clinical parameters (physical examination and vital signs, 12-lead electrocardiograms, documentation of adverse events, survival, review of CT scans of the tumour done clinically using response evaluation criteria in solid tumours–RECIST [35] and laboratory tests (haematology and blood chemistry). A fasting period for participants is not required for any test. All samples will be sent to the local hospital laboratory for analysis. Haematological parameters include full blood count (FBC) with differential count. Chemistry includes sodium, potassium, chloride, calcium, total protein, albumin and pre-albumin, aspartate aminotransferase, alanine transaminase, alkaline phosphatase, total bilirubin, creatinine (from which glomerular filtration rate will be calculated using the modification of diet in renal disease–MDRD formula; the current Australian national standard) [20], haemoglobin A1c (HbA1c) and random glucose. All participants with diabetes enrolled in this study will be required to monitor their blood glucose levels in consultation with their treating physician. Clinically meaningful increases in blood glucose levels will be reviewed as adverse events.

In addition, main clinical information including clinical diagnosis, histological or cytological tumour type will be detailed. The Charlson Comorbidity Index (CCI) will be used as a uniform measure of co-morbidity [18,19].

The participant diaries will form an important source of data collection. Information on compliance with study medication, step counts, food consumed for three complete days prior to each visit, unplanned healthcare contacts and 'out of pocket' costs will be recorded. Data will also be collected on caregivers' unmet needs, quality of life, and participants' and caregivers' direct care costs and resources use (Table 2).

## Statistical analysis

Statistical analysis will be undertaken using Statistical Package for the Social Sciences (SPSS) software, V28.0 (IBM Corporation, Armonk, NY; 2016). The primary analysis will concentrate on the feasibility metrics based on defined thresholds. As this is a feasibility trial, missing values will not be imputed.

**Table 2. Assessment methods and tools used in this study.**

| **Demographics and clinical information** |
|---|

- *Demographics*
- age and date of birth
  - biological gender
  - postcode
  - language spoken at birth and at home
  - ethnicity, including Aboriginal or Torres Strait Islander status
  - smoking history/ smoking status
  - country of birth
- *Clinical information*
  - main clinical diagnosis
  - histological or cytological tumour type, genotype and treatment
  - extent of disease will be recorded as limited or extensive for the purposes of stratification
  - small cell lung cancer treatment regimen and dose or interval adjustments
  - response to therapy for SCLC
  - adverse events from cancer therapy including nausea, vomiting, anorexia, fatigue, etc.
  - episodes of febrile neutropenia
  - date of death (including data of death from the time of enrolment)

| **Clinical and physical measurements** |
|---|

- *Height* in centimetres (cm) recorded to the nearest 0.5 cm while barefoot and standing.
- *Body weight* in kilograms (Kg).
- *Historical stable weight* by self-reported weight (Kg) in the last 1, 6 and 12 months.
- *Body mass index (BMI)* calculated from the patient's weight divided by the patient's height squared.
- *Body composition* obtained using:
  - dual x-ray absorptiometry (DEXA) with lean body mass and fat mass recorded;
  - bioelectrical impedance analysis (BIA);
  - muscle segmentation in axial computed tomography (CT) images at the thoracic (T4) level will be obtained from all electronically stored data from CT scans whenever they are done clinically as part of usual care. No new CT scans will be performed as part of this trial.
- *Complete physical examination* recorded in the source documentation at the site and any abnormalities at screening will be recorded in the AE eCRF.
- *Dedicated physical exam* aiming to monitor signs and symptoms of oedema such as pedal oedema, pericardial effusion, pleural effusions, and ascites.
- *Vital signs* including body temperature (˚C), blood pressure (mmHG, systolic and diastolic), respiratory rate (breaths per minute) and heart rate (beats per minute). Blood pressure and heart rate are measured after the patient has been resting in the semi-supine position for at least 5 minutes.
- *Complete medical history* including evaluations for past or present conditions, and concomitant medications.
- *Comorbidity assessment* recorder using the Charlson Comorbidity Index (without weighting) which incorporates in a single score the severity and number of comorbid conditions [18,19]. The index is an independent predictor of long-term survival (predicting the 10-year survival in patients with multiple co-morbidities).
- *12-lead ECGs* when possible, will be recorded at 1 hour (±5 minutes) post dose to align with occurrence of Cmax, except at screening. Any new or worsening ECG abnormalities assessed to be clinically significant by the study investigator will be reported as adverse events.
- *Short Blessed Test* at screening to assess cognitive capacity and will assist the PI to determine the persons capacity to understand the study and to complete the assessments. A score of 10 or more is required.
- *Adverse event assessment* the research team member who sees the participants and their caregivers will also ask about any other unexpected adverse outcomes.
- *Clinical Global Impression (CGI)* a two-item clinician-rated questionnaire designed to assess impression of disease severity (CGI-S) and global improvement or change (CGI-i) with a seven-point scale (1–7) with higher scores associated with greater severity and worsening.

| **Laboratory measurements** |
|---|

(*Continued*)

**Table 2.** (Continued)

- *Haematology*—full blood count (FBC) with differential count.
- *Blood chemistry*—sodium, potassium, chloride, calcium, total protein, albumin and pre-albumin, aspartate aminotransferase, alanine transaminase, alkaline phosphatase, total bilirubin, Creatinine (from which glomerular filtration rate will be calculated using the modification of diet in renal disease–MDRD formula; the current Australian national standard) [20], haemoglobin A1c (HbA1c) and random glucose.
- *Pregnancy screening*—for women of childbearing potential.
- *Cytokine biomarkers and level of infection/inflammation*—blood samples for IL-6 and C-reactive protein will be collected.

**Functional status and performance**

- *Australia—modified Karnofsky Performance Status (AKPS)*
  - validated variant of the Karnofsky Performance Status scale [21];
  - score of 0–100 (in increments of 10) indicating participant condition (in terms of physical ability) and can assist in prognostication;
  - higher scores suggest better condition.
- *Step counter and distance tracker*
  - a downloadable pedometer app available for free or a wearable digital pedometer (if participant do not have a mobile phone) will objectively track participant's physical activity (step count);
  - used at all times over 2 days (preferably 1 weekday and 1 weekend day immediately prior to visits);
  - daily step count to be entered on patient diary.
- *Timed up and go test (TUG)*
  - functional mobility test that requires the patient to stand up, walk 3 metres, turn, walk back and sit down [22];
  - presence of slowness, hesitancy, abnormal trunk or arm movements, staggering or stumbling is used to grade the patient from 1 (normal) to 5 (severely abnormal).

**Patient reported symptom measure**

- *Functional Assessment of Anorexia/Cachexia Therapy (FAACT)*
  - 12-item related to the patient's appetite in the last seven days [23];
  - 5-point Likert scale (zero equates to not at all and four equates to very much);
  - 5-item section referring to anorexia symptoms will be used to assess primary efficacy endpoint.
- *Global Impression of Change*
  - 7-point scale provides information about the participant perception of their change in clinical status, specifically their improvement since the commencement of the study [24];
  - from 'very much improved' to 'very much worse';
  - higher scores imply worse global impression of change.
- *EuroQol 5-Level 5-Dimensions (EQ-5D-5L)*
  - 5-question assessing health-related quality of life [25];
  - plus a VAS of current health-related quality of life, scored 0–100 [26–28]
- *ICECAP-A–ICEpop CAPability measure for Adults*
  - measure of capability for the general adult (18+) population for use in economic evaluation;
  - focuses on broader wellbeing, rather than health and comprises five attributes: attachment, stability, achievement, enjoyment and autonomy [29,30].
- *Short Questionnaire to Assess Health Enhancing Physical Activity–SQUASH*
  - provides an indication of the habitual activity level on an average week in the past month [31];
- *Functional Assessment of Chronic Illness Therapy-Fatigue (FACIT-F)*
  - measures the individual's level of fatigue during their usual daily activities over the past week [32]
  - 13-item with level of fatigue measured on a 4-point Likert scale
  - higher scores suggest less fatigue (range 0–52).
- *Participant acceptability of the study and proxy response measurement*
  - set of 3 items evaluating study acceptability at the time of participant's last visit;
  - information on required assistance to complete study tools evaluated by one screening question and two follow up questions.

**Participant diary**

(*Continued*)

**Table 2.** (Continued)

---

- **3-day food diary**
  - keep record of everything consumed for 3 complete days every 4–6 weeks (ideally 2 weekdays and 1 weekend day);
  - alternatively, a 24-h diet recall will be collected by the study nurse at the time of visit;
- **Record of daily step counts**
  - keep record of step counts for 2 complete days every 4–6 weeks (ideally 1 weekdays and 1 weekend day);
- **Record of compliance with study medication**
- **Unplanned healthcare costs**
  - resource use to estimate cost of indirect and informal care (e.g., time off work, out-of-pocket expenses, etc.)
  - unplanned healthcare contact/ hospitalisations defined as an emergency admission; or unplanned emergency contact with GP, nurses, specialists, home help visit, etc.; or unplanned readmission within 28 days following discharge to the same facility. All include as inpatient/ outpatient attendance and community care and participant may arrive at hospital in own transport or in an ambulance;
  - length of stay with primary reasons.

---

**Caregiver reported measure**

- **Carer Experience Scale (CES)**
  - profile measure focused on 'care-related quality of life' for use in economic evaluation [33,34].
  - comprises six attributes: activities, support from family and friends, assistance from organisations, fulfillment, control, and getting-on with the care recipient.
- **EuroQol 5-Level 5-Dimensions (EQ-5D-5L)**
  - 5-question assessing health-related quality of life [25];
  - plus a VAS of current health-related quality of life, scored 0–100 [26–28]
- **Caregiver diaries**
  - **Basic demographic and caregiver details**
    - assess changes in the caregiving needs.
  - **Unplanned healthcare costs**
    - resource use to estimate cost of indirect and informal care (e.g., time off work, caregiver time spend in-home nursing or accompanying to hospital, out-of-pocket expenses, etc.)

---

Descriptive statistics will be computed to describe the sample based on baseline demographic and clinical variables, and on feasibility outcomes. Categorical variables will be described using frequencies and proportion, and continuous valued variables will be summarised by their means, medians, standard deviations and interquartile ranges.

The efficacy outcomes will be analysed descriptively to inform the design of the future large, full-scale phase III study. The efficacy outcomes will be determined by calculating the difference between the baseline and Week 12 measures. The potential differences between arms will be performed using t-test for continuous normally distributed data or Wilcoxon Test for continuous non-normally distributed data. Additionally, linear mixed model for repeated measures with the efficacy outcomes measured at different time points as the dependent variables will be used to determine significance of change between the arms. Baseline efficacy outcome will be included as a covariate.

All safety and tolerability data will be summarised. Normally distributed data will be expressed as the mean ± standard deviation and analysed by an independent Student's t test. Non-normally distributed data will be reported as the median (interquartile range) and analysed using a Mann-Whitney U test. Categorical data will be expressed as the number of patients (percentage) and analysed using the chi-square test or Fisher's exact test. The datasets generated and analysed during the current study will be available from the corresponding author on reasonable request.

## Health economic analysis

Health economic analysis will determine indicative costs and benefits of anamorelin for treating anorexia in people with SCLC to the healthcare system and society; define and refine methods for data collection in a subsequent phase III clinical trial. Health economic analysis aims to provide answers to the following suite of questions: (i) are the costs of a full trial justified; (ii) how should costs and utility be measured as outcome measures, and (iii) how should the data be analysed and potentially modelled [36].

The following patient reported outcome measures will inform the analysis and include a generic preference-based QoL measure (EQ-5D-5L) [25–28] and the capability well-being measure (ICECAP-A) [29,30] elicited directly from patients. Carer QoL and unmet needs will be assessed using the EQ-5D-5L and Carer Experience Scale (CES) [33,34], respectively. Using a standard economic evaluation framework [37], we will determine the indicative cost of intervention relative to control. This information will be combined with the patient reported outcomes. The economic results will be presented from both a health system and societal perspective, with the latter also including impacts on carer time, productivity, and travel. Data will be sourced from the trial database, hospital system and diaries. Data collection schedule is provided in Fig 1. One way and probabilistic sensitivity analyses will be conducted.

## Confidentiality

Participants will be allocated a unique ID number. The master list linking identifying participant information and ID number will be maintained in a secure location, separate from the participant files and study database. Data collection worksheet and eCRF (electronic Case Report Form) tracking will use participant ID number only. There will be master lists held at each participating site. Study data will be stored and managed on an independent REDCap (Research Electronic Data Capture) database hosted by the University of Technology, Sydney (UTS). REDCap is a secure and password-protected web-based application where all web-based information transmission is encrypted and data are stored on a local UTS server with local management and support. All study data will be analysed by ID number only.

## Dissemination plans

Any requirements for data sharing during the process of publishing results in a peer-reviewed publication will comply with the data privacy and Human Research Ethics Committee requirements. Publications in scientific journals is planned as well as presentation of outcomes at national and international scientific meetings.

## Discussion

This study proposes to focus on anorexia in people with SCLC, as this symptom is rated among the top concerns (second only to fatigue) and there are currently limited treatment options [1]. Considering the unmet clinical need for safe and effective treatments for anorexia, anamorelin has the potential to improve anorexia and consequently, improve weight loss and nutritional status. This may lead to increased tolerance to anti-cancer therapies and better function maintenance, thus optimising quality of life for people with SCLC and, in the future, other cancers.

In this protocol, 100 mg was selected as the dose of anamorelin HCl. This dose had been evaluated in previously conducted Phase III studies (HT-ANAM-301 and HT-ANAM-302) [9,10] in NSCLC patients indicating that it was well-tolerated and showed improvement in cancer anorexia symptoms/concerns in people with NSCLC. A positive anabolic effect through both increased body weight and lean mass was also observed without androgenic side-effects.

In the safety extension study (HT-ANAM-303) [11], where participants from Studies HT-ANAM-301/302 continued dosing for up to an additional 12 weeks, a 100 mg dose level was also well tolerated, with no new safety signals identified in the longer dosing period. Due to an absence of a standard of care for anorexia that is tolerated with sustained benefit, a placebo arm is the chosen comparator arm.

The current safety profile of anamorelin was established from the ROMANA trials [9,11]. The most common drug-related treatment emergent harms observed were diabetes and hyperglycaemia, occurring in <1% of patients in ROMANA 1 and ROMANA 2 trials. Anamorelin also induced prolongation of PR and QRS intervals at supra-therapeutic doses of 300 and 400mg (based on 24-hour Holter monitoring data from Study HT-ANAM-112) [9,11]. The electrocardiogram (ECG) effects were then reassessed in a thorough QT/QTc study (Study HT-ANAM-113) [10,11] which confirmed that anamorelin had no meaningful effect on ventricular repolarisation. All ECG changes were transient, and none of the QT prolongations were above an absolute value of 450ms or were an increase of >60ms from baseline. Other associated harms were nausea, diarrhoea, peripheral oedema and fatigue. Considering these, this study will specifically seek evidence of cardiac dysfunction, symptomatic hyperglycaemia, drug interactions, constipation, nausea and vomiting using National Cancer Institute Common Terminology Criteria for Adverse Event (NCI-CTCAE) gradings. Importantly, this proposed trial is using a dose of 100mg which has not been clinically associated with any cardiac dysfunction.

The establishment of feasibility, desirability and the qualitative data from this study will inform the conduct of a subsequent robust effectiveness phase III trial. Meanwhile, the key primary and secondary efficacy clinical assessment outcomes will enable robust calculation of the sample size of a phase III trial of anamorelin versus placebo. Therefore, outcomes of this phase II study, if successful, will provide strong support for a full phase III trial using optimal measurable endpoints that will allow the most informative and clinically significant data to be obtained. In addition, a mixed-methods sub-study will help address patients' and caregivers' unmet needs as it will improve understanding of the conversations that occur during patient-clinician consultations regarding clinical history taking of anorexia, its burdens and impacts.

## Supporting information

**S1 Checklist. SPIRIT 2013 checklist: Recommended items to address in a clinical trial protocol and related documents\*.**
(DOC)

**S1 File.**
(PDF)

## Acknowledgments

The authors would like to acknowledge Ms Linda Brown, Mr John Stubbs and Dr Phillip Lee for their contributions to this clinical trial.

## Author Contributions

**Conceptualization:** Mariana S. Sousa, Peter Martin, Miriam J. Johnson, Michael Lind, Matthew Maddocks, Alex Bullock, Meera Agar, Sungwon Chang, Slavica Kochovska, Irina Kinchin, Victoria Bray, David C. Currow.

**Funding acquisition:** Mariana S. Sousa, Peter Martin, Miriam J. Johnson, Michael Lind, Matthew Maddocks, Alex Bullock, Sungwon Chang, Slavica Kochovska, Irina Kinchin, Victoria Bray, David C. Currow.

**Methodology:** Mariana S. Sousa, Peter Martin, Miriam J. Johnson, Michael Lind, Matthew Maddocks, Alex Bullock, Meera Agar, Sungwon Chang, Slavica Kochovska, Irina Kinchin, Deidre Morgan, Victoria Bray, David C. Currow.

**Project administration:** Mariana S. Sousa, David C. Currow.

**Supervision:** David C. Currow.

**Writing – original draft:** Mariana S. Sousa, Peter Martin, Miriam J. Johnson, Michael Lind, Matthew Maddocks, Alex Bullock, Meera Agar, Sungwon Chang, Slavica Kochovska, Irina Kinchin, Deidre Morgan, Belinda Fazekas, Valentina Razmovski-Naumovski, Jessica T. Lee, Malinda Itchins, Victoria Bray, David C. Currow.

**Writing – review & editing:** Mariana S. Sousa, Peter Martin, Miriam J. Johnson, Michael Lind, Matthew Maddocks, Alex Bullock, Meera Agar, Sungwon Chang, Slavica Kochovska, Irina Kinchin, Deidre Morgan, Belinda Fazekas, Valentina Razmovski-Naumovski, Jessica T. Lee, Malinda Itchins, Victoria Bray, David C. Currow.

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
