## [Decision Letter · Decision Letter 0]

1 Mar 2023

PONE-D-22-25611

Phase II, double blind, placebo controlled, multi-site study to evaluate the safety, feasibility and desirability of conducting a phase III study of anamorelin for anorexia in people with small cell lung cancer: a study protocol (LUANA trial)

PLOS ONE

Dear Dr. Currow,

Thank you for submitting your manuscript to PLOS ONE. After careful consideration, we feel that it has merit but does not fully meet PLOS ONE’s publication criteria as it currently stands. Therefore, we invite you to submit a revised version of the manuscript that addresses the points raised during the review process.

The reviewers feel that your manuscript does not sufficiently describe the plan for statistical analyses. As such, they recommend that you clearly detail all statistical methods that will be used to analyze the data. In addition, they feel that the manuscript requires additional information regarding the chosen exclusion criteria. Furthermore, they suggest that you should clarify the distinction between your study design and another ongoing clinical trial mentioned in your review (https://clinicaltrials.gov/ct2/show/NCT03743051). 

We look forward to receiving your revised manuscript.

Kind regards,

Alex Schaefer, PhD

Associate Editor

PLOS ONE

“This study is sponsored by the University of Technology Sydney and funded by the Helsinn Group including the provision of study medication and matching placebo. Helsinn will not dictate design, conduct, analysis, interpretation nor dissemination of the study’s findings. Helsinn was consulted on study design and will review the manuscript of the primary paper before final submission.  “

“Prof David Currow is a paid consultant to Helsinn and a member of their Advisory Board. All the other Authors declare that they have no conflict of interest.”

4. We note that the original protocol file you uploaded contains a confidentiality notice indicating that the protocol may not be shared publicly or be published. Please note, however, that the PLOS Editorial Policy requires that the original protocol be published alongside your manuscript in the event of acceptance. Please note that should your paper be accepted, all content including the protocol will be published under the Creative Commons Attribution (CC BY) 4.0 license, which means that it will be freely available online, and any third party is permitted to access, download, copy, distribute, and use these materials in any way, even commercially, with proper attribution.

Therefore, we ask that you please seek permission from the study sponsor or body imposing the restriction on sharing this document to publish this protocol under CC BY 4.0 if your work is accepted. We kindly ask that you upload a formal statement signed by an institutional representative clarifying whether you will be able to comply with this policy. Additionally, please upload a clean copy of the protocol with the confidentiality notice (and any copyrighted institutional logos or signatures) removed.

5. We note that the original protocol that you have uploaded as a Supporting Information file contains an institutional logo. As this logo is likely copyrighted, we ask that you please remove it from this file and upload an updated version upon resubmission.

Reviewers' comments:

Reviewer's Responses to Questions

**Comments to the Author**

1. Does the manuscript provide a valid rationale for the proposed study, with clearly identified and justified research questions?

Reviewer #1: Partly

Reviewer #2: Yes

2. Is the protocol technically sound and planned in a manner that will lead to a meaningful outcome and allow testing the stated hypotheses?

Reviewer #1: Partly

Reviewer #2: No

3. Is the methodology feasible and described in sufficient detail to allow the work to be replicable?

Reviewer #1: Yes

Reviewer #2: No

4. Have the authors described where all data underlying the findings will be made available when the study is complete?

Reviewer #1: No

Reviewer #2: No

5. Is the manuscript presented in an intelligible fashion and written in standard English?

Reviewer #1: Yes

Reviewer #2: Yes

6. Review Comments to the Author

You may also provide optional suggestions and comments to authors that they might find helpful in planning their study.

Reviewer #1: Mariana S. Sousa et al. Presented a Study Protocol LUANA trial, a randomized phase 2 trial with anamorelin for anorexia in people with small cell lung cancer.

My main concern is about the novelty of this trial. I would like to mentioned the fact that a previous trial using the design and the same drug is alrealdy ongoing in NSCLC: “NCT03743064 A Phase 3, randomized, double-blind, placebo-controlled, multicenter study to evaluate the efficacy and safety of anamorelin HCl for the treatment of malignancy associated weight loss and anorexia in adult patients with advanced non-small cell lung cancer (NSCLC).” So, the evaluation of the the safety, feasibility and desirability in LUANA trial sounds futile to me. The drugs used in SCLC with platinum-based chemotherapy associated with immunotherapy is also the backbone of NSCLC treatment . Therefore, ANAM 17-21 trail in NSCLC does already address LUANA phase II design presented in this work.

I am also concern about the fact the list of exclusion criteria:” Pleural effusion requiring thoracentesis. Pericardial effusion requiring drainage. Oedema requiring regular diuretics. Ascites requiring drainage. Oesophagitis, gastritis, Cognitive impairment….” Will exclude most of patient treated in daily routine with advanced SCLC which is a symptomatic disease related to high proportion of metastasis. Only “super Fit” patient with SLCLC and cachexia will be enrolled in the trial. I am quite sure that this kind of patient are highly selected population, consequently the accrual in the trial as well as the clinical application in daily routine practice is questionable.

Futhermore, this kind of trial should be a pantumor trial, unless if there is strong rational to support a specific tumor histology subtype or specific oncogenic driver.

Reviewer #2: In this study protocol, a multi-site, randomized-controlled phase II clinical trial aims to assess safety and feasibility, adherence, and completion of study tools to inform the design of a robust Phase III effectiveness trial.

Major revisions:

1- Overall, the current statistical analysis plan is too vague. List and describe all the statistical methods that will be used to analyze the data.

2- Line 355: Provide more details for the following statement. “Comparison between the intervention and control groups will occur using appropriate statistical approaches, depending on the nature of the data.”

3- Lines 359-362: A repeated measures approach would be superior to repeatedly applying paired t-tests.

Minor revisions:

1- Identify the software that will be used to capture the data as well as the software that will be used for the statistical analysis.

2- Line 355: Replace “analysed” with “summarized” and summarize using both frequencies and percentages.

7. PLOS authors have the option to publish the peer review history of their article (what does this mean?). If published, this will include your full peer review and any attached files.

Reviewer #1: No

Reviewer #2: No

---

## [Author Response · Author response to Decision Letter 0]

14 Apr 2023

Associate Editor Comments:

The reviewers feel that your manuscript does not sufficiently describe the plan for statistical analyses. As such, they recommend that you clearly detail all statistical methods that will be used to analyze the data. In addition, they feel that the manuscript requires additional information regarding the chosen exclusion criteria. Furthermore, they suggest that you should clarify the distinction between your study design and another ongoing clinical trial mentioned in your review (https://clinicaltrials.gov/ct2/show/NCT03743051). 

Response: Thank you for considering our paper for publication and for the suggestions to improve clarity. We have further described the plan for statistical analyses (please refer to pages 21 and 22 in the revised manuscript). We have also further clarified both the distinction between our design and ongoing trial, as well as about the chosen exclusion criteria (as per response to Reviewer’s 1 comments). 

Reviewers’ Comments to the Author’s:

Reviewer #1: 

Mariana S. Sousa et al. Presented a Study Protocol LUANA trial, a randomized phase 2 trial with anamorelin for anorexia in people with small cell lung cancer.

My main concern is about the novelty of this trial. I would like to mentioned the fact that a previous trial using the design and the same drug is alrealdy ongoing in NSCLC: “NCT03743064 A Phase 3, randomized, double-blind, placebo-controlled, multicenter study to evaluate the efficacy and safety of anamorelin HCl for the treatment of malignancy associated weight loss and anorexia in adult patients with advanced non-small cell lung cancer (NSCLC).” So, the evaluation of the the safety, feasibility and desirability in LUANA trial sounds futile to me. The drugs used in SCLC with platinum-based chemotherapy associated with immunotherapy is also the backbone of NSCLC treatment. Therefore, ANAM 17-21 trail in NSCLC does already address LUANA phase II design presented in this work.

Response:

We acknowledge Reviewer’s #1 concerns and wanted to reiterate that this is a feasibility study and this trial is trying to shift the way we analyse data on anamorelin which is primarily an appetite enhancer. Currently, there is no standard of care for cachexia/anorexia and no new drug has been approved by the FDA in the USA, EMA in Europe and TGA in Australia, thus remains an unmet clinical need. While these regulatory agencies have indicated in the past that clinical trials testing new drugs for cachexia/anorexia had to demonstrate meaningful improvements in muscle mass and function as co-primary endpoints, currently this is not necessarily the case due to lack of international consensus on cachexia/anorexia endpoints. Furthermore, promising results for anamorelin as an appetite stimulant in previous clinical trials have been reported which suggests its benefit for anorexia. Therefore, the rationale for the selection of our primary endpoints was based on ongoing developments in the field. 

The above mentioned phase 3 trial focuses on people with both NSCLC-associated weight loss (determined as BMI<20kg/m2 with involuntary weight loss of >2% within 6 months prior to screening) and anorexia (determined as ongoing problems with appetite/eating associated with the underlying cancer, as determined by having score of ≤ 17 points on the 5-item Anorexia Symptom Scale (5-IASS) and ≤ 37 points on the 12-item FAACT A/CS). The primary outcome measures in this trial are mean change in body weight and mean change in the 5-IASS. Our trial on the other hand, was designed to demonstrate the superiority of anamorelin versus placebo on anorexia symptoms and the ability to complete therapy on time and on dose.

Cont. Reviewer #1: 

I am also concern about the fact the list of exclusion criteria:” Pleural effusion requiring thoracentesis. Pericardial effusion requiring drainage. Oedema requiring regular diuretics. Ascites requiring drainage. Oesophagitis, gastritis, Cognitive impairment….” Will exclude most of patient treated in daily routine with advanced SCLC which is a symptomatic disease related to high proportion of metastasis. Only “super Fit” patient with SLCLC and cachexia will be enrolled in the trial. I am quite sure that this kind of patient are highly selected population, consequently the accrual in the trial as well as the clinical application in daily routine practice is questionable.

Response:

Similar to our trial, the exclusion criteria for the NCT03743064 Phase 3 trial also includes pleural effusion requiring thoracentesis, pericardial effusion requiring drainage, oedema requiring regular diuretics, ascites requiring drainage, oesophagitis, gastritis, cognitive impairment, etc. The reason for the selection of such exclusion criteria was based on previously reported data that underlined the development of the Investigator’s Brochure for anamorelin and the fact that any oral and gastrointestinal impairments will impact on the patient’s ability to eat which is the desired effect of anamorelin. While there are very small implications for people experiencing these symptoms, they cannot be ignored within a trial design. For instance, as identified adverse events in clinical trials (HT-ANAM-301, HT-ANAM-302, HT-ANAM-303, RC-1291-203, RC-1291-205, RC-1291-206, ST-ANAM-207), pleural effusion and pericardial effusion were present in 2.6% and 1.6% of participants, respectively.

Cont. Reviewer #1: 

Furthermore, this kind of trial should be a pantumor trial, unless if there is strong rational to support a specific tumor histology subtype or specific oncogenic driver.

Response: While we acknowledge your suggestion, the feasibility of including a diverse (pantumoural) population would make it difficult to achieve homogeneity of the sample. We are looking for signals and trends in the data and understand that by adding different variables and confounders would limit our ability to confidently drawn any conclusions. We have for this reason focused on SCLC which is a population largely affected by anorexia. We also acknowledge that NSCLC and pancreatic cancer are the main populations studies with this medication to date.

Reviewer #2: 

In this study protocol, a multi-site, randomized-controlled phase II clinical trial aims to assess safety and feasibility, adherence, and completion of study tools to inform the design of a robust Phase III effectiveness trial.

Major revisions:

1- Overall, the current statistical analysis plan is too vague. List and describe all the statistical methods that will be used to analyze the data.

Response: Thank you for your suggestions. We have improved substantially the analysis section of the manuscript and included a detailed statistical plan. Please refer to pages 21 and 22 in the manuscript. 

2- Line 355: Provide more details for the following statement. “Comparison between the intervention and control groups will occur using appropriate statistical approaches, depending on the nature of the data.”

Response: We have revised the statistical plan and description in the manuscript. Please refer to pages 21 and 22. 

3- Lines 359-362: A repeated measures approach would be superior to repeatedly applying paired t-tests.

Response: Thank you for your suggestions. We have improved substantially the analysis section of the manuscript and included a detailed statistical plan. Please refer to pages 21 and 22 in the manuscript. 

Minor revisions:

1- Identify the software that will be used to capture the data as well as the software that will be used for the statistical analysis.

Response: We have included the information about the software that will be used to capture the data and for statistical analysis on page 21, lines 353-354 in the manuscript. 

2- Line 355: Replace “analysed” with “summarized” and summarize using both frequencies and percentages.

Response: We have revised this information in the manuscript, please refer to page 21 lines 357- 368.

---

## [Decision Letter · Decision Letter 1]

3 May 2023

Phase II, double blind, placebo controlled, multi-site study to evaluate the safety, feasibility and desirability of conducting a phase III study of anamorelin for anorexia in people with small cell lung cancer: a study protocol (LUANA trial)

PONE-D-22-25611R1

Dear Dr. Currow,

We’re pleased to inform you that your manuscript has been judged scientifically suitable for publication and will be formally accepted for publication once it meets all outstanding technical requirements.

Kind regards,

Junichi Matsubara

Academic Editor

PLOS ONE

Additional Editor Comments (optional):

Reviewers' comments:

Reviewer's Responses to Questions

**Comments to the Author**

1. Does the manuscript provide a valid rationale for the proposed study, with clearly identified and justified research questions?

Reviewer #1: Partly

Reviewer #2: Yes

2. Is the protocol technically sound and planned in a manner that will lead to a meaningful outcome and allow testing the stated hypotheses?

Reviewer #1: No

Reviewer #2: Yes

3. Is the methodology feasible and described in sufficient detail to allow the work to be replicable?

Reviewer #1: No

Reviewer #2: Yes

4. Have the authors described where all data underlying the findings will be made available when the study is complete?

Reviewer #1: Yes

Reviewer #2: No

5. Is the manuscript presented in an intelligible fashion and written in standard English?

Reviewer #1: Yes

Reviewer #2: Yes

6. Review Comments to the Author

You may also provide optional suggestions and comments to authors that they might find helpful in planning their study.

Reviewer #1: I do believe that The primary outcome measure in LUANA trial: mean change in body weight and mean change in the 5-IASS are the good readouts in the context of cachexia/anorexia in cancer. The design used here to demonstrate the superiority of anamorelin versus placebo on anorexia symptoms and the ability to complete therapy on time and on dose is less strong than weight gain or change in the 5-IASS.

Futhermore, Olanzapine improves chemo-related anorexia and has become un standard of care to prevent nausea. Therefore control arm without olanzapine is questionable.

Regarding exclusion criteria: first of all the NCT03743064 Phase 3 trial has been targeted NSCLC. We know that SCLC is the most agressive lung cancer commonly associted with pleural , pericardial effusion. Additionaly , gastritis and oesophagitis is a quite comme disease whith impact in oral, food intake in only severe cases. So i do strongly believe that these exclusions do represent the majority of SCLC patients. The exclusions criteria should fit those from the trial below.

Randomized Double-Blind Placebo-Controlled Study of Olanzapine for Chemotherapy-Related Anorexia in Patients With Locally Advanced or Metastatic Gastric, Hepato-pancreatico-biliary, and Lung Cancer.

Sandhya L. et al.

J Clin Oncol . 2023 Mar 28;JCO2201997. doi: 10.1200/JCO.22.01997. Online ahead of print

Reviewer #2: All comments have been adequately addressed.

7. PLOS authors have the option to publish the peer review history of their article (what does this mean?). If published, this will include your full peer review and any attached files.

Reviewer #1: **Yes: **Frank Aboubakar

Reviewer #2: No

---

## [Editor Report · Acceptance letter]

8 May 2023

PONE-D-22-25611R1 

Phase II, double blind, placebo controlled, multi-site study to evaluate the safety, feasibility and desirability of conducting a phase III study of anamorelin for anorexia in people with small cell lung cancer: a study protocol (LUANA trial) 

Dear Dr. Currow:

I'm pleased to inform you that your manuscript has been deemed suitable for publication in PLOS ONE. Congratulations! Your manuscript is now with our production department. 

Kind regards, 

on behalf of

Dr. Junichi Matsubara 

Academic Editor

PLOS ONE